# A microtranslatome coordinately regulates sodium and potassium currents in the human heart

Catherine A Eichel, Erick B Ríos-Pérez, Fang Liu, Margaret B Jameson, David K Jones[†], Jennifer J Knickelbine, Gail A Robertson*

Department of Neuroscience and Cardiovascular Research Center, School of Medicine and Public Health, University of Wisconsin-Madison, Madison, United States

**Abstract** Catastrophic arrhythmias and sudden cardiac death can occur with even a small imbalance between inward sodium currents and outward potassium currents, but mechanisms establishing this critical balance are not understood. Here, we show that mRNA transcripts encoding $I_{Na}$ and $I_{Kr}$ channels (*SCN5A* and *hERG*, respectively) are associated in defined complexes during protein translation. Using biochemical, electrophysiological and single-molecule fluorescence localization approaches, we find that roughly half the *hERG* translational complexes contain *SCN5A* transcripts. Moreover, the transcripts are regulated in a way that alters functional expression of both channels at the membrane. Association and coordinate regulation of transcripts in discrete 'microtranslatomes' represents a new paradigm controlling electrical activity in heart and other excitable tissues.
DOI: https://doi.org/10.7554/eLife.52654.001

*For correspondence: garobert@wisc.edu

Present address: [†]Department of Pharmacology, University of Michigan Medical School, Ann Arbor, United States

Competing interests: The authors declare that no competing interests exist.

## Introduction

Signaling in excitable cells depends on the coordinated flow of inward and outward currents through a defined ensemble of ion channel species. This is especially true in heart, where the expression of many different ion channels controls the spread of excitation triggering the concerted contraction of the ventricular myocardium. Even small perturbations in the quantitative balance due to block or mutations affecting a single type of channel can initiate or perpetuate arrhythmias and lead to sudden death. Repolarization is a particularly vulnerable phase of the cardiac cycle, when imbalance of inward and outward currents can prolong action potential duration and trigger arrhythmias such as Torsades de Pointes (*Roden, 2016*). The genetic basis of such catastrophic arrhythmias is in many cases unknown; mechanisms coordinating expression of multiple ion channels may represent novel disease targets.

Cardiac $I_{Kr}$ is critical for normal repolarization (*Sanguinetti and Jurkiewicz, 1990*) and is a major target of acquired and congenital long QT syndrome (*Sanguinetti et al., 1995*; *Trudeau et al., 1995*). $I_{Kr}$ channels minimally comprise hERG1a and hERG1b subunits (*Sale et al., 2008*; *Jones et al., 2004*), which associate cotranslationally (*Phartiyal et al., 2007*) and preferentially form heteromultimers (*McNally et al., 2017*). Underlying heteromultimerization is the cotranslational association of *hERG1a* and *1b* mRNA transcripts (*Liu et al., 2016*). Because current magnitude is greater in heteromeric hERG1a/1b vs. homomeric hERG1a channels, and loss of hERG1b is pro-arrhythmic (*Sale et al., 2008*; *Jones et al., 2014*), the mechanism of cotranslational assembly of hERG subunits is important in cardiac repolarization (*Liu et al., 2016*).

In this study we found that association of transcripts could occur not only between alternate *hERG* transcripts encoded by a single gene locus, but also between transcripts encoding entirely

different ion channel types whose balance is critical to cardiac excitability. Indeed, we show that *SCN5A*, encoding the cardiac Na$_v$1.5 sodium channel, associates with *hERG* transcripts as demonstrated by co-immunoprecipitation of nascent protein in heterologous expression systems, cardiomyocytes derived from human induced pluripotent stem cells, and native human myocardium. Single-molecule fluorescent in situ hybridization (smFISH) quantitatively reveals *hERG* and *SCN5A* transcript colocalization captured during protein translation. Targeting *hERG* transcripts for shRNA degradation coordinately reduces *SCN5A* transcript levels as well, along with native $I_{Kr}$ and $I_{Na}$ currents recorded from cardiomyocytes. Thus, cotranslational association and regulation of transcripts is a novel mechanism establishing and preserving a balance of $I_{Kr}$ and $I_{Na}$ in heart, where relative levels of these currents critically determine normal action potential production and coordinated electrical activity.

## Results

### Copurification of *hERG1a* and *SCN5A* transcripts with their encoded proteins

Using specific antibodies that target the N-terminus of hERG1a, we purified hERG1a protein from induced pluripotent stem cell-derived cardiomyocytes (iPSC-CMs) and human ventricle lysates and performed RT-PCR to identify associated transcripts ('RNA-IP'; *Figure 1A*). As previously reported (*Liu et al., 2016*), both *hERG1a* and *hERG1b* transcripts co-immunoprecipitated with nascent hERG1a protein. Surprisingly, *SCN5A* transcripts encoding Na$_v$1.5 channels also copurified with nascent hERG1a protein (*Figure 1B* and *Figure 1—figure supplement 1*). The interaction appears specific since neither ryanodine receptor RyR2 nor inward rectifier channel Kir2.1 (*KCNJ2*) transcripts copurified as part of this complex. The counterpart experiment using anti-Na$_v$1.5 antibodies confirmed association of transcripts encoding hERG1a, hERG1b and Na$_v$1.5, but not RyR2 (*Figure 1B*). Bead-only controls showed no signal, indicating specific interactions of antibodies with corresponding antigens. The association also occurred in HEK293 cells, where additional controls showed that the antibodies used did not interact nonspecifically with mRNA encoding the other ion channels or subunits (*Figure 1—figure supplement 1*). Interestingly, when lysates independently expressing hERG1a and Na$_v$1.5 were mixed, hERG1a antibodies copurified only *hERG1a* mRNA, and Nav1.5 antibodies copurified only *SCN5A* mRNA, indicating that association of the two mRNAs requires their co-expression in situ. In addition, the interaction between *hERG1a* and *SCN5A* does not require the presence of *hERG1b* (*Figure 1—figure supplement 1*). This experiment demonstrates that transcripts encoding hERG1a, hERG1b and Na$_v$1.5 physically interact within the cell and can be copurified using antibodies targeting either hERG1a or Na$_v$1.5 nascent proteins. Their association with either encoded protein implies the transcripts associate during protein translation, or *cotranslationally*.

### *hERG1a* and *SCN5A* transcript distribution

To independently confirm *hERG1a* and *SCN5A* transcript association, we performed single-molecule fluorescence in situ hybridization (smFISH) experiments in iPSC-CMs (*Figure 2A*). We used a combination of short DNA oligonucleotides (20 nucleotides), each labeled with a single fluorophore, that bind in series on the target mRNA and collectively are detected as a single fluorescent spot (*Raj et al., 2008*) (see Materials and methods). Probes for *hERG1a* and *SCN5A* mRNAs were designed with spectrally separable labels for simultaneous detection (Quasar 647 and 546 respectively; see Materials and methods and *Figure 2—figure supplement 1* for probe validation, and *Supplementary file 1* for list of probes) (*Femino et al., 1998*). Punctate signal for each mRNA species appeared singly and in clusters (*Figure 2A–B*). To evaluate mRNA copy number in each detected signal, we fitted the histogram of the total fluorescence intensity of smFISH signals with the sum of Gaussian functions and determined mean intensity of a single mRNA molecule for each species (*Figure 2B*; *Figure 2—figure supplements 2–3*). We found that approximately 25% of detected molecules exist singly, whereas about 20% occupy clusters containing six or more transcripts (*Figure 2C*). Both transcripts were observed throughout the cytoplasm with higher density within 5–10 µm from the nucleus (*Figure 2A and D*), consistent with the expected distribution of perinuclear endoplasmic reticulum where these mRNA molecules are translated into proteins. A

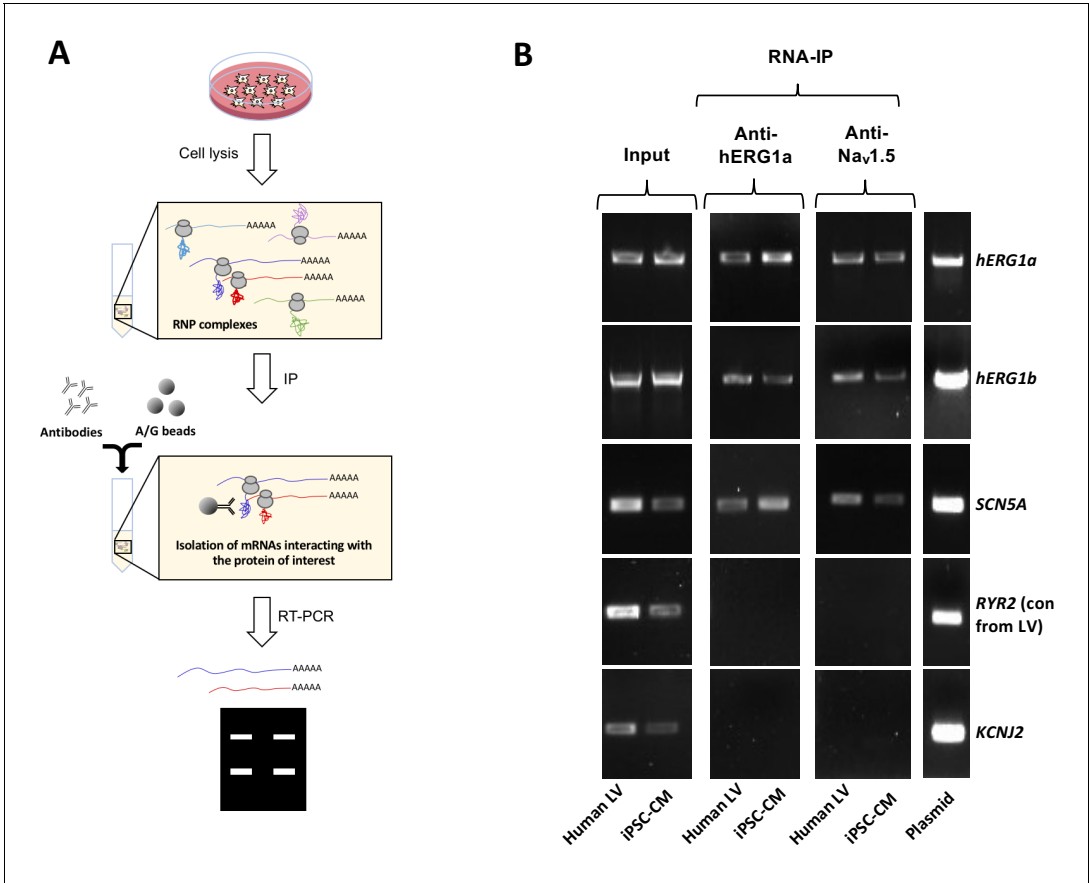

**Figure 1.** Complex of ion channel transcripts with nascent proteins. (**A**) Scheme of the RNA-IP protocol in which channel-specific antibodies are used to pull down nascent proteins and associated transcripts. RNP: ribonucleoprotein. (**B**) *Lanes 1 and 2*, RT-PCR products from input lysate of human left ventricle (LV), and iPSC-CM. *Lanes 3–16* shows the corresponding RNA-IP's using an anti-hERG1a or anti-Na$_v$1.5 antibodies; Lane seven shows the control (+) and represents signal amplified from purified plasmid template. Similar results were obtained in at least three independent experiments. (N = 5 for anti-hERG1a and N = 3 for anti-Nav1.5 using human LV and iPSC-CMs).

DOI: https://doi.org/10.7554/eLife.52654.002

The following source data and figure supplement are available for figure 1:

**Source data 1.** RNA-IP Blots raw data for *Figure 1B*.
DOI: https://doi.org/10.7554/eLife.52654.004
**Figure supplement 1.** Complete RNA-IP from *Figure 1*.
DOI: https://doi.org/10.7554/eLife.52654.003

*GAPDH* mRNA probe set served as a positive control for smFISH experiments (Stellaris validated control). In contrast with signals observed for *hERG1a* and *SCN5A* transcripts, *GAPDH* transcript clustered less, with 50% found as single molecules and <5% in clusters of 6 or more transcripts (*Figure 2C*). Moreover, *GAPDH* molecules distributed more homogeneously throughout the cytoplasm with higher density in the range of 10 to 20 µm from the nucleus (*Figure 2D*). We noted similar numbers of *hERG1a* and *SCN5A* transcripts per cell but fewer than those for *GAPDH* (*Figure 2E*). Thus, numbers and spatial distribution of *hERG1a* and *SCN5A* transcripts can be simultaneously resolved. Further work will be required to elucidate the significance or possible physiological role of differently sized mRNA clusters.

## *hERG1a* and *SCN5A* transcript expression levels correlate

Although we observed a range in numbers of *hERG1a* and *SCN5A* mRNAs among iPSC-CMs (*Figure 2E*), regression analysis revealed clear correlation in their expression levels within a given cell (*Figure 3* and *Supplementary file 2*). Plotted against each other, *hERG1a* and *SCN5A* mRNA numbers exhibited a coefficient of determination ($R^2$) of 0.57 (p=0.00001; 41 cells; *Figure 3A–B*). In

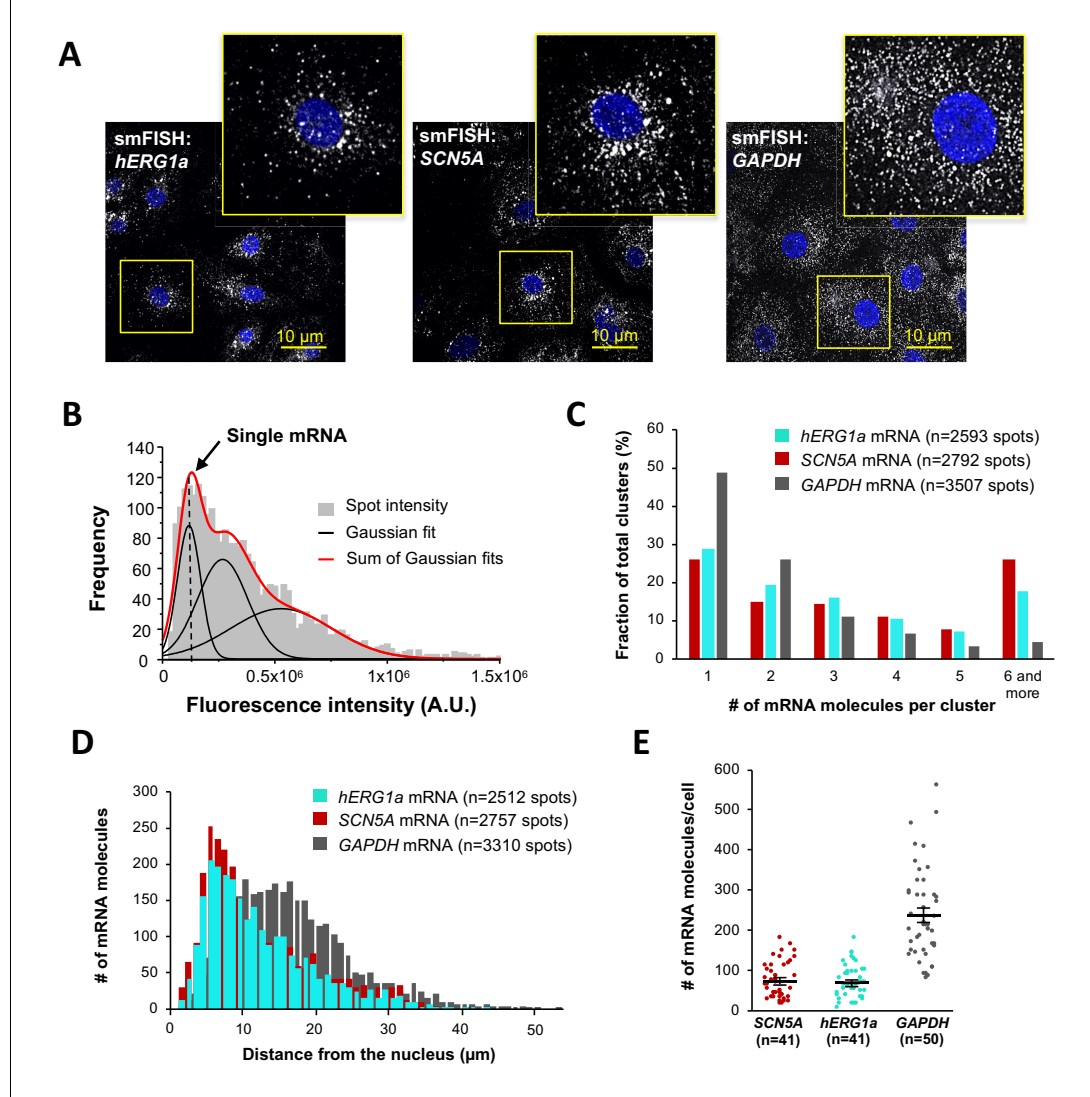

**Figure 2.** Quantitative description of single *hERG1a* and *SCN5* transcripts and their distribution in iPSC-CMs. (**A**) Representative confocal images and enlargement (outlined in yellow) of iPSC-CMs subjected to the smFISH protocol. (**B**) By fitting the intensity histogram of smFISH signals (n = 2611 spots) to the sum of Gaussian functions (red line), the typical intensity corresponding to a single mRNA molecule (vertical dashed line) was extracted. (**C**) The distribution of the number of mRNA molecules associated in clusters for each transcript evaluated by smFISH. (**D**) Histogram showing the cytoplasmic distribution of mRNA signals with distance from the nucleus. (**E**) The number of mRNAs detected per cell was plotted for *SCN5A, hERG1a* and *GAPDH* (lines represent mean ±SE).

DOI: https://doi.org/10.7554/eLife.52654.005

The following source data and figure supplements are available for figure 2:

**Source data 1.** Intensities plot for determination of single mRNA intensity raw data for *Figure 2B*.
DOI: https://doi.org/10.7554/eLife.52654.010
**Source data 2.** Clusterization of transcripts raw data for *Figure 2C*.
DOI: https://doi.org/10.7554/eLife.52654.011
**Source data 3.** Distance of mRNA from the nucleus raw data for *Figure 2D*.
DOI: https://doi.org/10.7554/eLife.52654.012
**Source data 4.** Numbers of mRNA per cells raw data for *Figure 2E*.
DOI: https://doi.org/10.7554/eLife.52654.013
**Figure supplement 1.** Specificity of the probes used in smFISH experiments.
DOI: https://doi.org/10.7554/eLife.52654.006
**Figure supplement 2.** Single mRNA intensity determination.
DOI: https://doi.org/10.7554/eLife.52654.007

*Figure 2 continued on next page*

*Figure 2 continued*

**Figure supplement 2—source data 1.** Single mRNA intensity determination for *Figure 2—figure supplement 2*.

DOI: https://doi.org/10.7554/eLife.52654.008

**Figure supplement 3.** Quantification of mRNA expression using two different methods.

DOI: https://doi.org/10.7554/eLife.52654.009

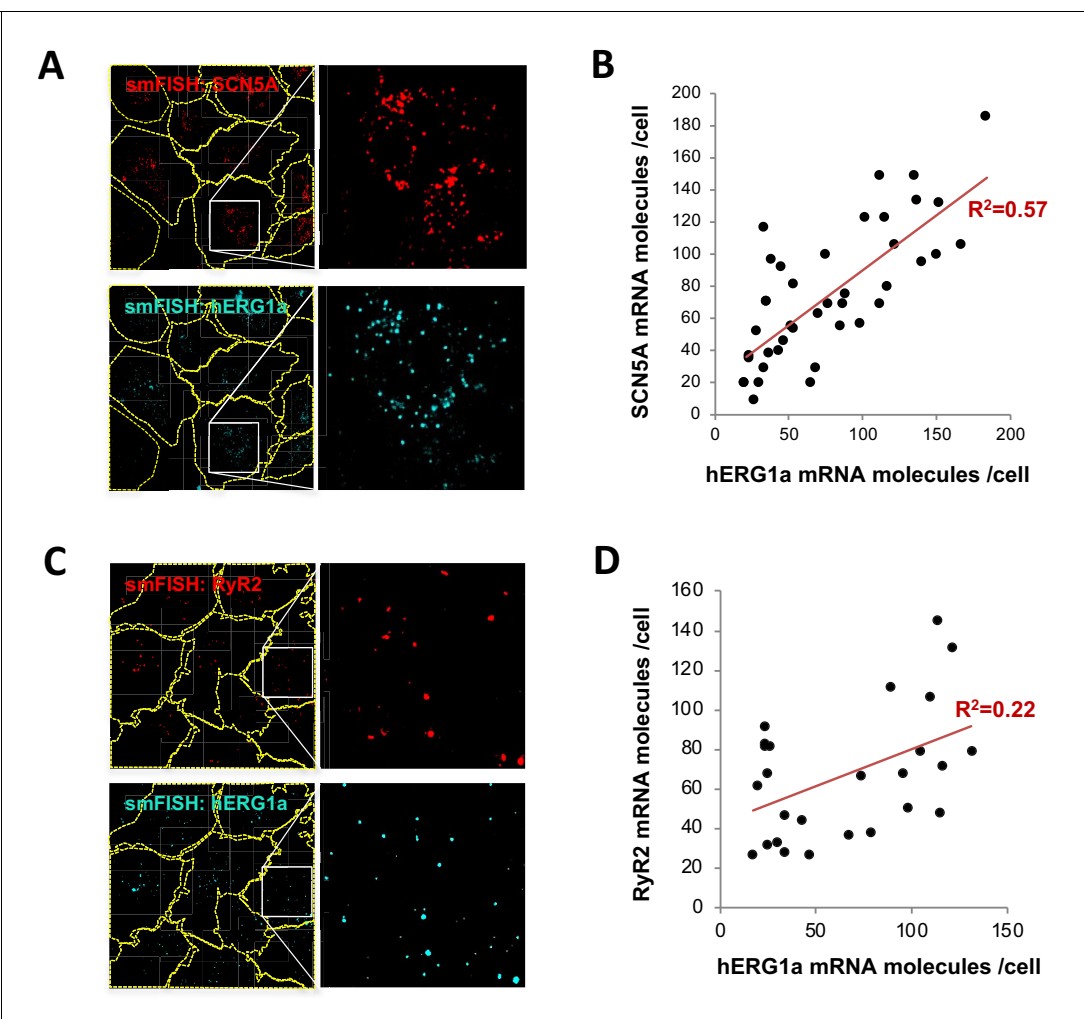

**Figure 3.** *hERG1a* and *SCN5A* transcript expression levels correlate. (**A**) Representative confocal images and enlargements of double smFISH experiments for *SCN5A* (red) and *hERG1a* (cyan) mRNAs. (**B**) The number of mRNA molecules detected per cell in double smFISH experiments were plotted for *SCN5A* and *hERG1a* and the coefficient of determination $R^2$ was determined from the Pearson's correlation coefficient R (n = 41 cells; N = 2). (**C**) Representative confocal images and enlargements of double smFISH experiments for *RyR2* (red) and *hERG1a* (cyan) mRNAs. (**D**) The number of *hERG1a* mRNA was plotted against the number of *RYR2* mRNAs per cells and showed a low correlation in their expression (n = 26 cells; N = 2).

DOI: https://doi.org/10.7554/eLife.52654.014

The following source data and figure supplements are available for figure 3:

**Source data 1.** Correlation analysis of *hERG1a* and *SCN5A* mRNA expressions raw data for *Figure 3B*.

DOI: https://doi.org/10.7554/eLife.52654.017

**Source data 2.** Correlation analysis of *hERG1a* and *RyR2* mRNA expressions raw data for *Figure 3D*.

DOI: https://doi.org/10.7554/eLife.52654.018

**Figure supplement 1.** Correlation of mRNA expression.

DOI: https://doi.org/10.7554/eLife.52654.015

**Figure supplement 1—source data 1.** Correlation of mRNA expression for *Figure 3—figure supplement 1*.

DOI: https://doi.org/10.7554/eLife.52654.016

contrast, pairwise combinations of *hERG1a* and *RyR2*, *hERG1a* and *GAPDH*, or *SCN5A* and *GAPDH* exhibited much lower linear correlation ($R^2$ = 0.22, p=0.017; $R^2$ = 0.18, p=0.15; and $R^2$ = 0.33, p=0.000134 respectively; n = 26, 13, and 28 cells respectively; *Figure 3C–D*, *Figure 3—figure supplement 1A–B*, and *Supplementary file 2*). Spearman coefficients revealed similar results as Pearson coefficients, where significant correlation is observed only between *SCN5A* and *hERG1a* (*Supplementary file 2*). These findings indicate a roughly constant ratio of *hERG1a* and *SCN5A* mRNA copies.

## *hERG1a* and *SCN5A* transcripts colocalize

To determine potential *hERG1a* and *SCN5A* transcript association using smFISH, we measured proximity between the two signals using the centroid position, scored from touching to 67% (1 pixel) overlap (*Figure 4A–B*). To discern colocalization from random overlap, we calculated the expected number of particles that could associate based on chance only for the different association criteria. Two-tailed *t* tests with Bonferroni correction revealed association between *hERG1a* and *SCN5A* transcripts significantly greater than that expected by chance (see Materials and methods; P values summarized in *Supplementary file 3*; *Figure 4B*). Approximately 25% of each transcript population was associated with the other (*Figure 4C*). To test specificity of interaction between *hERG1a* and *SCN5A* transcripts, smFISH and pairwise comparisons were also performed with *RyR2* and *GAPDH* transcripts, which revealed no significant association (*Figure 4D–E*; *Supplementary file 3*). These results show that association of *hERG* and *SCN5A* transcripts demonstrated in lysates can also be visualized in iPSC-CMs in situ, and provide strong evidence for the existence of a discrete mRNA complex comprising *hERG1a* and *SCN5A* transcripts.

## Discrete *hERG1a* and *SCN5A* cotranslational complexes

To further explore whether colocalized mRNAs were part of a translational complex, we combined smFISH with immunofluorescence using hERG1a antibodies. We observed close association between *hERG1a* and *SCN5A* mRNAs and hERG1a protein significantly greater than that expected by chance (*Figure 5A–B* and *Figure 5—figure supplement 1A–B*). Interestingly, among the 16% of actively translated *hERG1a* mRNAs (i.e. those associated with hERG1a protein), 46% were also associated with *SCN5A* mRNAs (*Figure 5C*), indicating a 3-fold enrichment of their association in translational complexes. Analysis of the distribution of colocalized molecules revealed that 70% are located close to the nucleus (within 10 μm, *Figure 5D*).

We monitored association of hERG1a protein and transcript in the presence of puromycin, which releases translating ribosomes from mRNAs (*Azzam and Algranati, 1973*) (*Figure 6A*). We observed no change due to puromycin in the total number of respective mRNAs detected per cell (*Figure 6B*). As expected, puromycin reduced association between *hERG1a* mRNA and hERG1a protein (antibody) and the S6 ribosomal protein (*Figure 6C*). In addition, triple colocalization of *hERG1a* and *SCN5A* transcripts and either hERG1a protein or the ribosomal subunit S6 was robustly reduced (*Figure 6D*). These findings further support the conclusion that *hERG1a* and *SCN5A* associate cotranslationally.

## *hERG1a* and *SCN5A* mRNAs are coregulated

We previously demonstrated that targeted knockdown of either *hERG1a* or *1b* transcripts by specific short hairpin RNA (shRNA) caused a reduction of both transcripts not attributable to off-target effects in iPSC-CMs or HEK293 cells (*Liu et al., 2016*). To determine whether *hERG* and *SCN5A* transcripts are similarly subject to this co-knockdown effect, we evaluated expression levels by performing RT-qPCR experiments in iPSC-CMs. We found that *hERG1a*, *hERG1b* and *SCN5A* expression levels were all reduced by about 50% upon *hERG1a* silencing compared to the effects of a scrambled shRNA (*Figure 7A*, orange bars). *RYR2* transcript levels were unaffected. We observed similar results using the specific hERG1b shRNA (*Figure 7A*, blue bars). Expressed independently in HEK293 cells, only *hERG1a* mRNA was affected by the 1a shRNA, and only *hERG1b* was affected by the 1b shRNA (*Figure 7B*). *SCN5A* was unaffected by either shRNA, indicating that the knockdown in iPSC-CMs was not due to off-target effects and levels of associated *hERG1a* and *SCN5A* are quantitatively coregulated. Similar results of approximately 40% co-knockdown of discrete *hERG1a* and *SCN5A* mRNA particles were obtained using smFISH (*Figure 7—figure supplement 1*). Even more

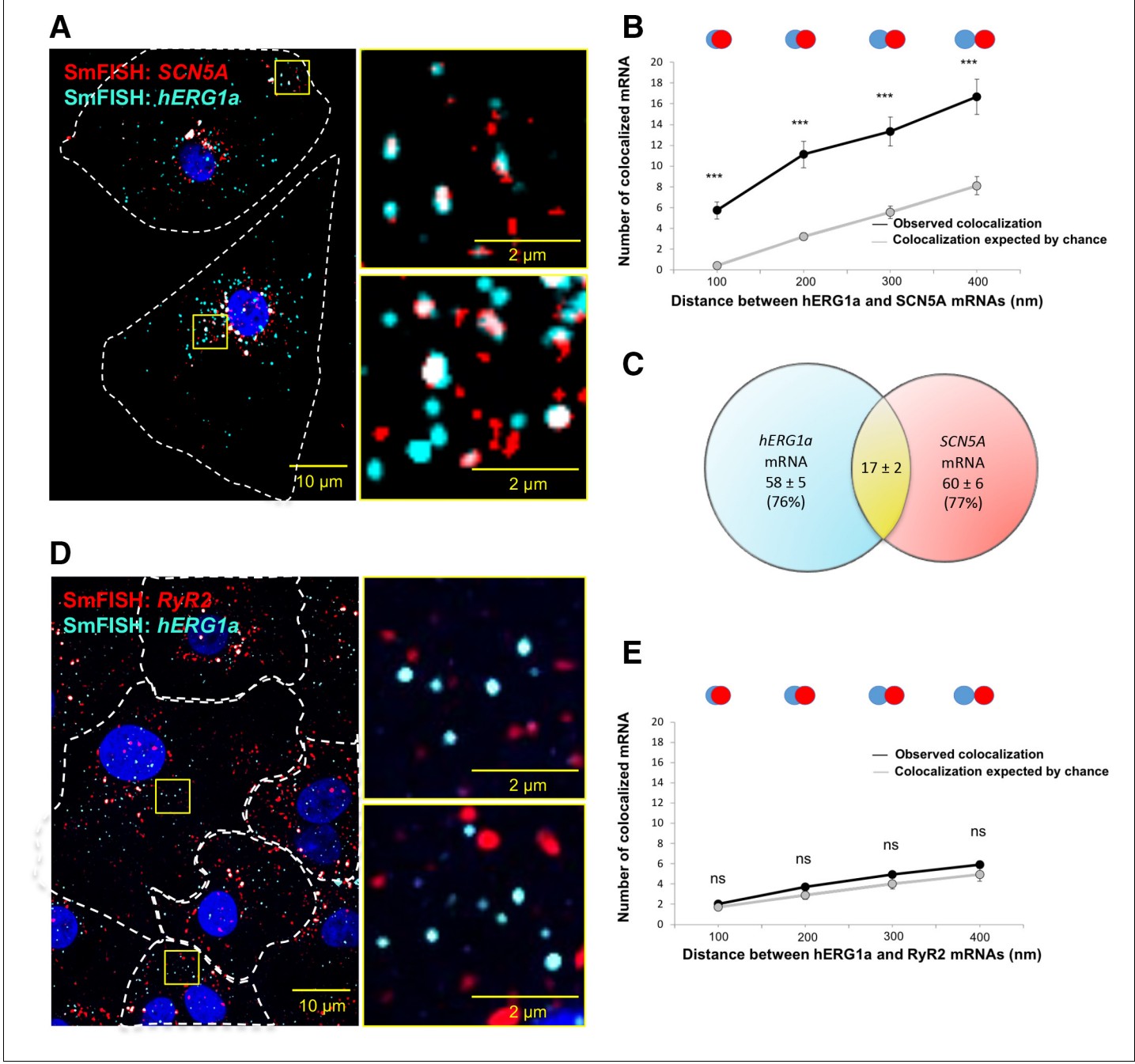

**Figure 4.** hERG1a and SCN5a transcript colocalization. (**A**) Representative confocal images and enlargement (outlined in yellow) of iPSC-CMs subjected to smFISH showing the colocalization of *hERG1a* and *SCN5A* mRNAs. (**B**) Comparison of the average number of associated *hERG1a* and *SCN5A* mRNAs particles observed vs. expected by chance using different overlap criteria illustrated (mean ±SE; n = 41 cells; N = 2). (**C**) Diagram illustrating that the association of *hERG1a* and *SCN5A* mRNAs account for 24% and 23% of their total population respectively. (**D**) Representative confocal images of smFISH for hERG1a and RyR2 transcripts. (**E**) Comparison of the average number of associated *hERG1a* and *RyR2* mRNAs particles observed vs. expected by chance using different overlap criteria (mean ±SE; n = 26 cells; N = 2).

DOI: https://doi.org/10.7554/eLife.52654.019

The following source data is available for figure 4:

**Source data 1.** Association of *hERG1a* and *SCN5A* transcripts raw data for *Figure 4B*.
DOI: https://doi.org/10.7554/eLife.52654.020

**Source data 2.** Proportion of *hERG1a* and *SCN5A* mRNA association raw data for *Figure 4C*.
DOI: https://doi.org/10.7554/eLife.52654.021

*Figure 4 continued on next page*

*Figure 4 continued*

**Source data 3.** Association of *hERG1a* and *RyR2* transcripts raw data for *Figure 4E*.
DOI: https://doi.org/10.7554/eLife.52654.022

than the total population of mRNA, the number of colocalized particles is decreased by approximately 55%, indicating that physically associated transcripts are subjected to co-knockdown (*Figure 7—figure supplement 1C*). Together these results indicate a coordinated and quantitative regulation of mRNAs encoding a complement of ion channels.

## $I_{Kr}$ and $I_{Na}$ are coregulated

To assess functional consequences of transcript coregulation, we recorded effects of *hERG1b* silencing on native currents in iPSC-CMs. *Figure 7C* shows the repolarizing current $I_{Kr}$ in iPSC-CMs transfected with either hERG1b or scrambled shRNA. Steady state and peak tail $I_{Kr}$ were decreased in *hERG1b*-silenced cells compared to cells transfected with scrambled shRNA (*Figure 7D*). $I_{Kr}$ reduction was the result of a decrease in $G_{max}$ upon hERG1b-specific silencing with no modifications in the voltage dependence of activation (*Figure 7E* and *Supplementary file 4*). These results are in accordance to our previous studies reporting a reduction in $I_{Kr}$ density upon *hERG1b*-specific silencing, and indicate that transcripts targeted by shRNA are those undergoing translation (*Liu et al., 2016*; *Jones et al., 2014*). To determine whether *hERG1b* silencing also affects translationally active *SCN5A*, we measured peak $I_{Na}$ density in iPSC-CMs and detected significant reduction of about 60% when *hERG1b* was silenced, compared to control cells (*Figure 7F–H*). Peak $G_{max}$ was decreased but no alterations in voltage dependence of activation or inactivation were detected (*Figure 7H* and *Supplementary files 4* and *5*). Late $I_{Na}$, measured as the current integral from 50 to 800 ms from the beginning of the pulse (*Glynn et al., 2015*), was similarly reduced in magnitude (*Figure 7I–K*). This analysis indicates that coregulation via co-knockdown results in quantitatively similar alteration of $I_{Na,late}$ and $I_{Kr}$, which operate together to regulate repolarization (*Banyasz et al., 2011*). $I_{to}$, which does not regulate action potential duration in larger mammals (*Sun and Wang, 2005*), is unaffected by *hERG1b* silencing (*Figure 8A–D*), suggesting the coregulation of $I_{Na}$ and $I_{Kr}$ reflects their coherent participation in repolarization.

## Discussion

We have demonstrated using diverse and independent approaches the association and coregulation of transcripts encoding ion channels that regulate excitability in cardiomyocytes. By co-immunoprecipitating mRNA transcripts along with their nascent proteins, we have shown that *hERG* and *SCN5A* transcripts associate natively in human ventricular myocardium and iPSC-CMs as well as when heterologously expressed in HEK293 cells. Using smFISH together with immunofluorescence in iPSC-CMs, we demonstrate that the ratio of *hERG* and *SCN5A* transcripts is approximately 1:1 despite a range of pool sizes from roughly 5 to 200 molecules per cell. These transcripts colocalize about 25% of the time, but when considering only those *hERG* transcripts undergoing translation, nearly 50% are associated with *SCN5A*. When *hERG1a* or *hERG1b* transcripts are targeted by shRNA, *SCN5A* levels are reduced by about the same amount. Both peak and late $I_{Na}$ are correspondingly reduced. Reflecting their coherent roles in the process of cardiac repolarization, the term 'microtranslatome' captures the cotranslational properties of this discrete complex comprising functionally related mRNAs and their nascent proteins.

What is the functional role of cotranslational association of transcripts? Deutsch and colleagues showed that cotranslational interaction of nascent Kv1.3 N-termini facilitates proper tertiary and quaternary structure required for oligomerization (*Tu and Deutsch, 1999*; *Robinson and Deutsch, 2005*). Cotranslational heteromeric association of hERG1a and hERG1b subunits ensures cardiac $I_{Kr}$ has the appropriate biophysical properties and magnitude shaping the normal ventricular action potential. Coordinated protein translation of *different* channel types could control relative numbers of ion channels involved in electrical signaling events. Such a balance is critical during repolarization, when alterations in $I_{Kr}$ or late $I_{Na}$ are known to cause arrhythmias associated with long QT syndrome or Brugada syndrome (*Rook et al., 1999*; *Bezzina et al., 1999*; *Bennett et al., 1995*). Indeed,

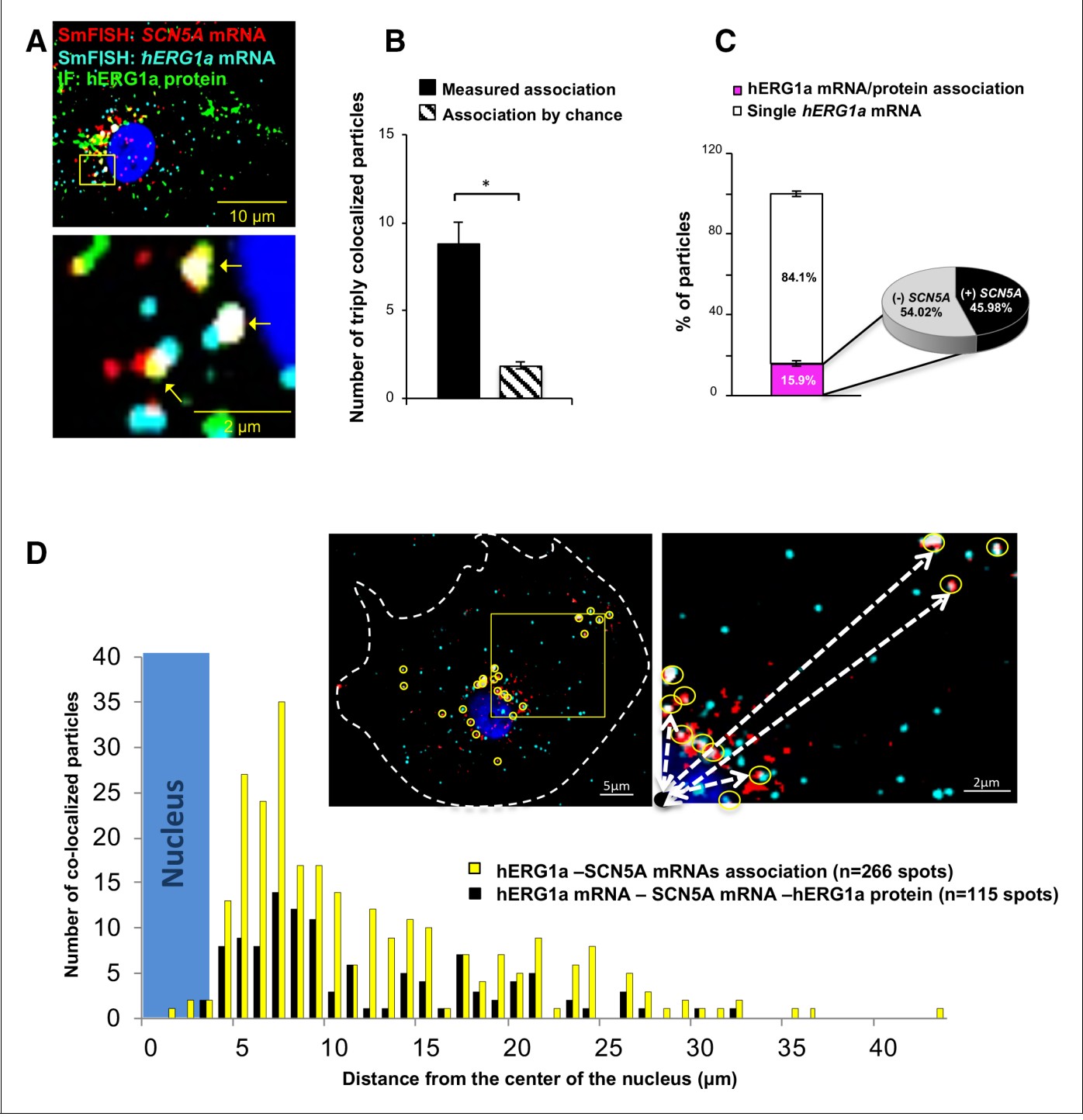

**Figure 5.** Cotranslational association of hERG1a protein and *hERG1a* and *SCN5A* mRNAs. (**A**) Representative confocal images and enlargement of iPSC-CMs subjected to immunofluorescence (IF) combined with smFISH protocol. Arrows indicate triply colocalized particles. (**B**) The average number of particles comprising *hERG1a* and *SCN5A* mRNAs and hERG1a protein per cell compared to the expected number based on chance using a maximum distance of 2 pixels between center of mass (minimum 50% overlap; mean ±SE; n = 13 cells, N = 2). (**C**) Histogram showing that 16% of *hERG1a* mRNA associate with hERG1a protein (actively translated population); of that percentage, 46% also interact with *SCN5A* transcripts (mean ±SE; n = 13 cells; N = 2). (**D**) Histogram showing the distribution of colocalized mRNA spots through the cytoplasm and from the nucleus revealing that RNP complexes are mostly localized within 10 μm from the nucleus. In the top right corner, representative examples of colocalized spots (yellow circles) and analysis of distance from the nucleus (white dashed arrows).

DOI: https://doi.org/10.7554/eLife.52654.023

*Figure 5 continued on next page*

*Figure 5 continued*

The following source data and figure supplements are available for figure 5:

**Source data 1.** Association of *hERG1a* and *SCN5A* mRNAs with hERG1a protein raw data for *Figure 5B*.
DOI: https://doi.org/10.7554/eLife.52654.026
**Source data 2.** Proportion of co-translational association raw data for *Figure 5C*.
DOI: https://doi.org/10.7554/eLife.52654.027
**Source data 3.** Distribution of associated mRNAs raw data for *Figure 5D*.
DOI: https://doi.org/10.7554/eLife.52654.028
**Figure supplement 1.** hERG1a mRNA protein interaction.
DOI: https://doi.org/10.7554/eLife.52654.024
**Figure supplement 1—source data 1.** Association of hERG1a and SCN5A mRNAs with hERG1a protein raw data for *Figure 5—figure supplement 1*.
DOI: https://doi.org/10.7554/eLife.52654.025

during normal Phase 3 repolarization, non-equilibrium gating of sodium channels leads to recovery from inactivation and re-activation of currents substantially larger than the tiny steady-state late $I_{Na}$ observed under voltage-clamp steps (*Banyasz et al., 2011*; *Clancy et al., 2003*). Our observation of roughly equivalent *hERG1a* and *SCN5A* mRNA levels squares with previous reports of fixed channel transcript ratios associated with certain identified crustacean neurons (*Schulz et al., 2007*; *Schulz et al., 2006*). Cotranslating mRNAs in a stoichiometric manner could buffer noise associated with transcription (*Dar et al., 2012*) and render a stable balance of channel protein underlying control of membrane potential.

These studies raise questions of the mechanism by which transcripts associate. Although hERG1a and hERG1b N-termini interact during translation (*Phartiyal et al., 2007*), association of transcripts does not rely on this interaction: alternate transcripts encoding the proteins interact even when translation of one of the proteins is prevented (*Liu et al., 2016*). In principle, transcripts could associate via complementary base pairing or by tertiary structural interactions as ligand and receptor. Alternatively, they could be linked by one or more RNA binding proteins (RBPs). Because the association and coregulation observed in native heart can be reproduced in HEK293 cells, the same or similar mechanisms are at work in the two systems. More work will be required to discern among possible mechanisms, and to determine the time course with respect to transcription, nuclear export and cytosolic localization of interacting transcripts.

A mechanism involving RBPs is appealing because it comports with the idea of the 'RNA regulon,' a term describing a complex of transcripts bound by one or more RBPs (*Brown et al., 2001*; *Keene and Tenenbaum, 2002*). RBPs in the yeast Puf family bind large collections of mRNAs to control their localization, stability, translation and decay (*Gerber et al., 2004*; *García-Rodríguez et al., 2007*). In mammalian systems, the Nova protein serves to coordinate expression of mRNAs encoding splicing proteins important in synaptic function (*Ule et al., 2003*). Presumably in both cases these proteins interact in multiple regulons (complexes) serving different or related roles. Mata and colleagues isolated individual mRNA species in yeast and showed they associate with other mRNAs encoding functionally related (but nonhomologous) proteins, along with mRNA encoding the RBP itself (*Duncan and Mata, 2011*). Moreover, these mRNAs encoded proteins that formed stable macromolecular complexes (*Duncan and Mata, 2014*). Taking it one step further, *Cosker et al. (2016)* showed that two mRNAs involved in cytoskeletal regulation bind the same RBP to form a single RNA granule, possibly analogous to the microtranslatome regulating key elements of excitability in the heart reported here.

A comprehensive analysis of the microtranslatome's components will require RNA-seq at a level of multiplexing that ensures sufficient statistical power in the face of potentially reduced complexity of the RNA-IP samples. These efforts will necessarily be followed by validation through complementary approaches such as RNAi and smFISH to confirm their identity within the microtranslatome.

One of the more curious findings of our study is the coordinate knockdown of different mRNAs in the complex by shRNAs targeted to only one of the mRNA species. The mechanism by which multiple mRNA species may be simultaneously regulated is not clear. shRNAs silence gene expression by producing an antisense (guide) strand that directs the RNA-induced silencing complex (RISC) to cleave, or suppress translation of, the target mRNA (*Petersen et al., 2006*; *Maroney et al., 2006*). Since hERG shRNA has no off-target effect on *SCN5A* mRNA expressed heterologously in HEK293

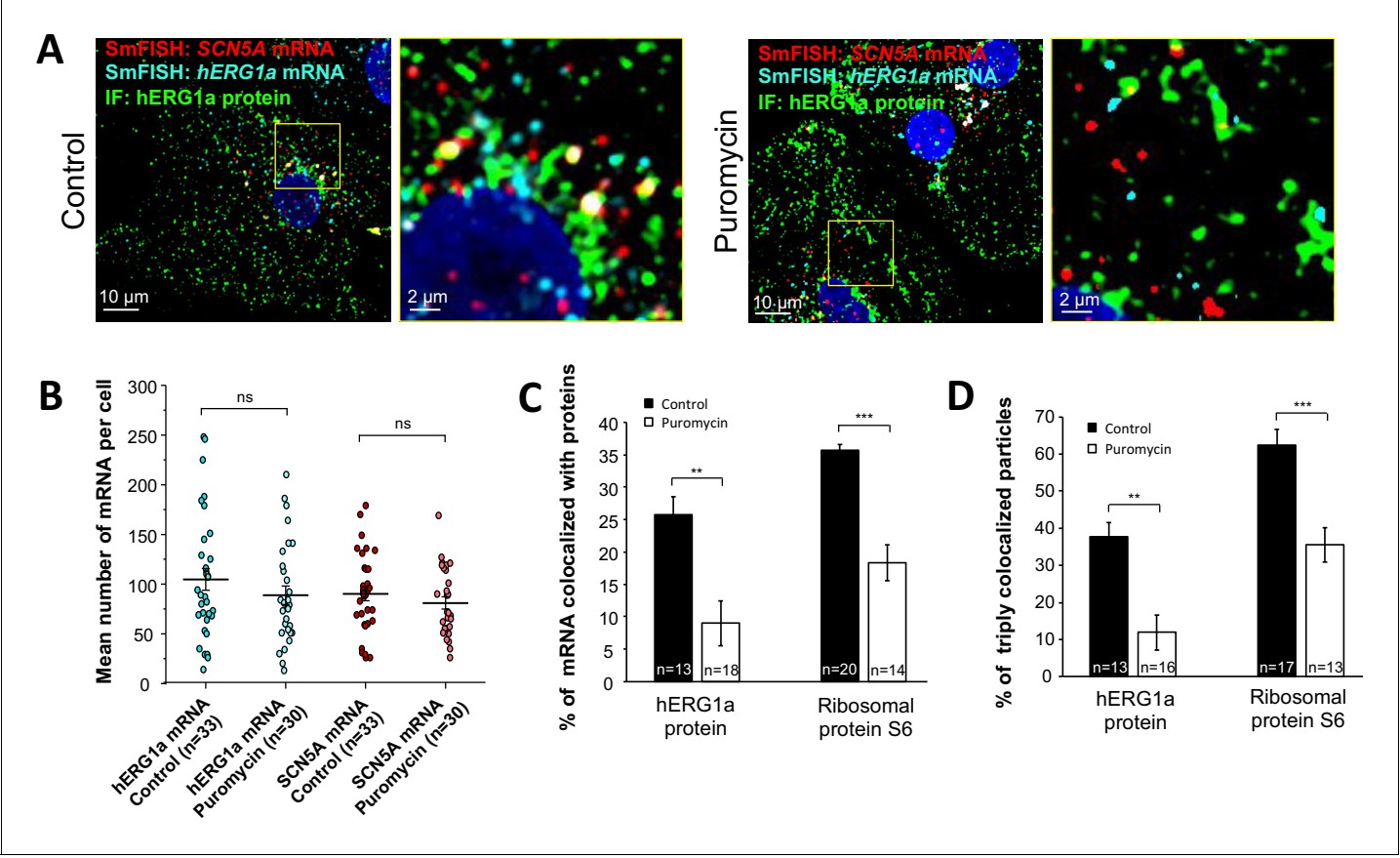

**Figure 6.** Distribution and association of *hERG1a* and *SCN5A* transcripts under puromycin treatment in iPSC-CMs. (**A**) Representative confocal images and enlargement (outlined in yellow) of iPSC-CMs subjected to immunofluorescence combined with smFISH for control cells (left panel) or cells treated with 100 µM puromycin for 15 min (right panel). (**B**) The number of mRNAs detected per cell was plotted for *SCN5A and hERG1a* in the presence of puromycin and compared to control cells (lines represent mean ±SE). (**C**) Histogram showing the reduction of association between *hERG1a* mRNA and hERG1a protein after puromycin treatment compared to non-treated cells (mean ±SE). (**D**) Histogram showing that the % of triply colocalized particles (hERG1a protein or the ribosomal subunit S6 associated with both *hERG1a* and *SCN5A* mRNAs) is decreased upon puromycin treatment (mean ±SE).

DOI: https://doi.org/10.7554/eLife.52654.029

The following source data is available for figure 6:

**Source data 1.** Number of mRNA per cell after puromycin raw data for *Figure 6B*.
DOI: https://doi.org/10.7554/eLife.52654.030
**Source data 2.** *hERG1a* and *SCN5A* mRNAs association raw data for *Figure 6C*.
DOI: https://doi.org/10.7554/eLife.52654.031
**Source data 3.** *hERG1a* and *SCN5A* transcripts cotranslational association raw data for *Figure 6D*.
DOI: https://doi.org/10.7554/eLife.52654.032

cells, we assume there is insufficient complementarity for a direct action. Perhaps by proximity to RISC, translation of the nontargeted mRNA is also disrupted, but to our knowledge no current evidence is available to support this idea. A transcriptional feedback mechanism seems unlikely given that co-knockdown can occur with plasmids transiently expressed from engineered promoters and not integrated into the genome of HEK293 cells. It is also important to note that it is unknown whether *SCN5A* is the only sodium channel transcript coregulated by *hERG* knockdown. In principle, transcripts encoding other sodium channels implicated in late $I_{Na}$, such as Nav1.8 (*Yang et al., 2012*; *Macri et al., 2018*), could also be affected, as could transcripts encoding auxiliary subunits associated with Nav1.5 (*Isom et al., 1994*).

Whether disrupting the integrity of these complexes gives rise to some of the many arrhythmias not attributable to mutations in ion channel genes *per se* remains to be determined. Although the coregulation of inward $I_{Na}$ and outward $I_{Kr}$ shown in this study may suggest a compensatory

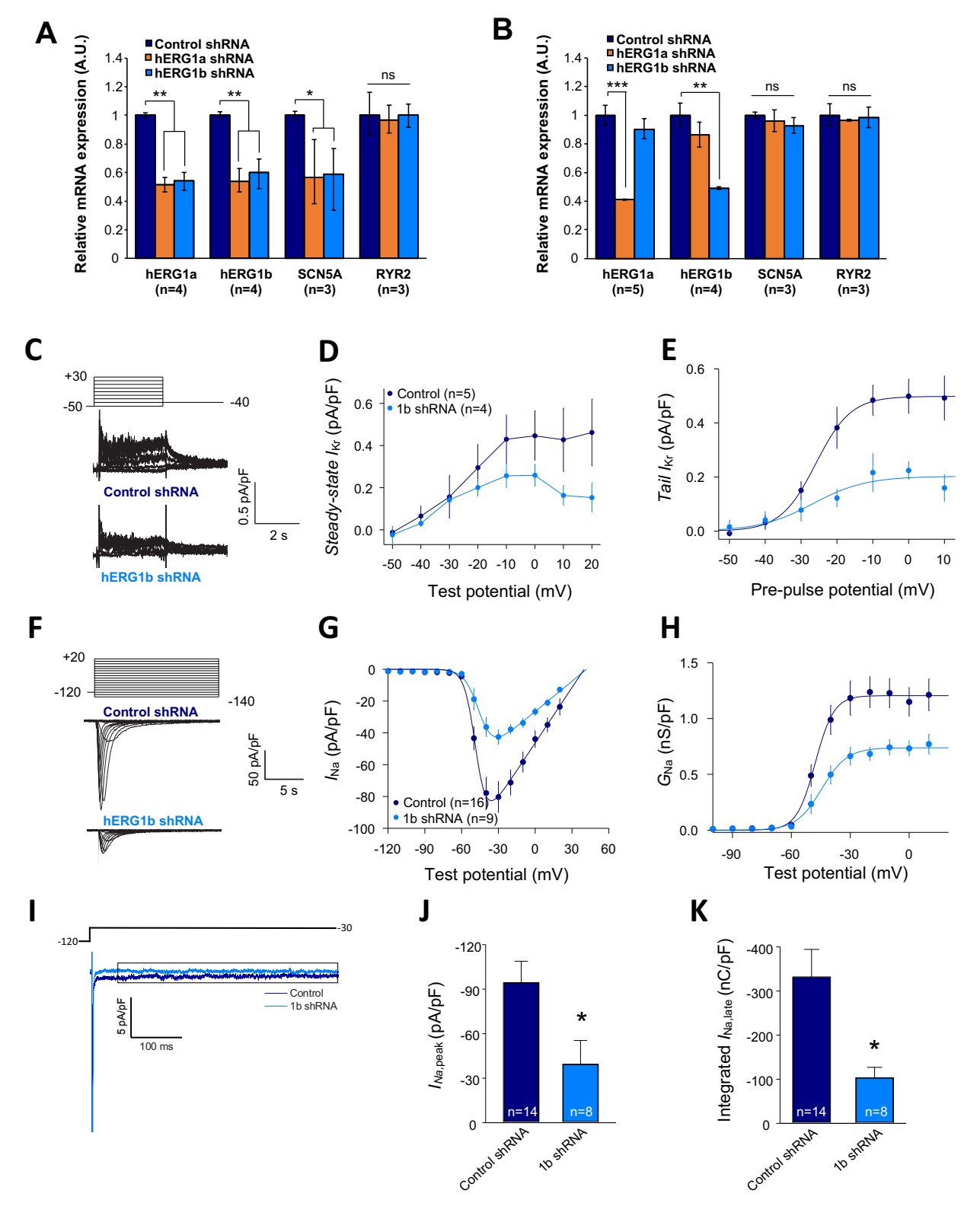

**Figure 7.** Co-knockdown of $I_{Kr}$ and $I_{Na}$ by *hERG* transcript-specific shRNA. (A) Effects of *hERG1a* or *hERG1b* silencing on channel mRNA expression levels detected by RT-qPCR (mean ±95% CI) in iPSC-CMs. A non-targeting shRNA (scrambled shRNA) is used as a control. (B) Effects of specific *hERG1a* or *hERG1b* silencing on ion channel mRNAs expressed alone in HEK293 cells. (C) Representative family of traces show $I_{Kr}$ in presence of control (upper) or hERG1b shRNA (lower). (D) Summary of steady-state current density vs. test potential shows effect of hERG1b shRNA (mean ±SE). (E) Effects

*Figure 7 continued on next page*

*Figure 7 continued*

of 1b shRNA on peak tail current vs. pre-pulse potential (mean ±SE). (**F**) Representative family of traces recorded from iPSC-CMs showing effects of hERG1b-specific shRNA compared to control shRNA on peak $I_{Na}$. (**G**) Summary current-voltage plot of peak $I_{Na}$ vs. test potential (mean ±SE). (**H**) Summary conductance (G)-voltage plot based on data from **G** (mean ±SE). (**I**) Late sodium current representative trace in control and 1b shRNA-transfected cells, measured by applying a single pulse protocol of 800 ms. (**J**) Summary statistics of peak $I_{Na}$ showed a decrease upon transfection with hERG1b shRNA (mean ±SE). (**K**) Late $I_{Na}$ measured as the integral from 50 to 800 ms from the beginning of the pulse showed a decrease upon transfection with hERG1b shRNA (mean ±SE).

DOI: https://doi.org/10.7554/eLife.52654.033

The following source data and figure supplements are available for figure 7:

**Source data 1.** Co-knockdown of transcripts by qPCR raw data for *Figure 7A*.

DOI: https://doi.org/10.7554/eLife.52654.036

**Source data 2.** Specificity of shRNA raw data for *Figure 7B*.

DOI: https://doi.org/10.7554/eLife.52654.037

**Source data 3.** $I_{Kr}$ is reduced upon hERG silencing raw data for *Figure 7D–E*.

DOI: https://doi.org/10.7554/eLife.52654.038

**Source data 4.** Reduction of peak $I_{Na}$ after hERG silencing raw data for *Figure 7G–H*.

DOI: https://doi.org/10.7554/eLife.52654.039

**Source data 5.** Decrease of $I_{Na,late}$ current upon hERG silencing raw data for *Figure 7J–K*.

DOI: https://doi.org/10.7554/eLife.52654.040

**Figure supplement 1.** Co-knockdown of *hERG* and *SCN5A* mRNAs by hERG transcript-specific shRNA.

DOI: https://doi.org/10.7554/eLife.52654.034

**Figure supplement 1—source data 1.** Co-knockdown of hERG and SCN5A by hERG transcript-specific shRNA for *Figure 7—figure supplement 1*.

DOI: https://doi.org/10.7554/eLife.52654.035

mechanism, in a previous study we showed that selective knockdown of *hERG1b* prolongs action potential duration and enhances variability, both cellular markers of proarrhythmia (*Jones et al., 2014*). Perhaps in the absence of co-regulation the effects would be more deleterious. Jalife and colleagues have introduced the concept of the 'channelosome,' a macromolecular protein complex mediating a physiological action. Interestingly, Nav1.5 and Kir2.1, which regulates resting and diastolic membrane potential, exhibit compensatory changes when the levels of either are genetically manipulated (*Milstein et al., 2012*). In this case, the effect seems to be on stability of the nontargeted channel proteins, which form a complex together with SAP97, and not on mRNA levels (*Matamoros et al., 2016*). We do not yet know whether the complex of transcripts we have studied encodes a similarly stable macromolecular complex, or perhaps ensures appropriate ratios of channels distributed independently at the membrane. Based on current evidence, we propose that the microtranslatome of associated transcripts is a novel mechanism governing the quantitative expression of multiple ion channel types and thus the balance of excitability in the cardiomyocyte.

## Materials and methods

### Cell lines, culture, plasmids and transfection

HEK293 cells were purchased from ATCC and cultured under standard conditions (37°C, 5% $CO_2$) in DMEM medium (Gibco) supplemented with 10% Fetal Bovine Serum (FBS, Gibco). iPSC-CMs (iCell, Cellular Dynamics International) were plated and cultured following manufacturer's instructions. A certificate of analysis including purity and identity, sterility, mycoplasma absence, plating efficiency and viability is provided with each vial. We performed additional mycoplasma testing after plating in the laboratory. ShRNA sequences specific for hERG1a 5'-GCGCAGCGGCTTGCTCAACTCCACC TCGG-3' and its control 5'-GCACTACCAGAGCTAACTCAGATAGTACT-3' were provided by Origene into a pGFP-V-RS vector. shRNA specific for hERG1b 5'-CCACAACCACCCTGGCTTCAT-3' and its respective control were purchased from Sigma-Aldrich. For heterologous expression, hERG1a (NM_000238) and hERG1b (NM_172057) sequences were cloned into pcDNA3.1. Transient transfections were performed using 2.5 µl/ml Lipofectamin 2000 (Thermofisher) with 2 µg/ml plasmid. Cells were collected for further analysis 48 hr after transfection. When needed, a second transfection was performed 24 hr after the first one with either hERG1a or hERG1b shRNA and the corresponding

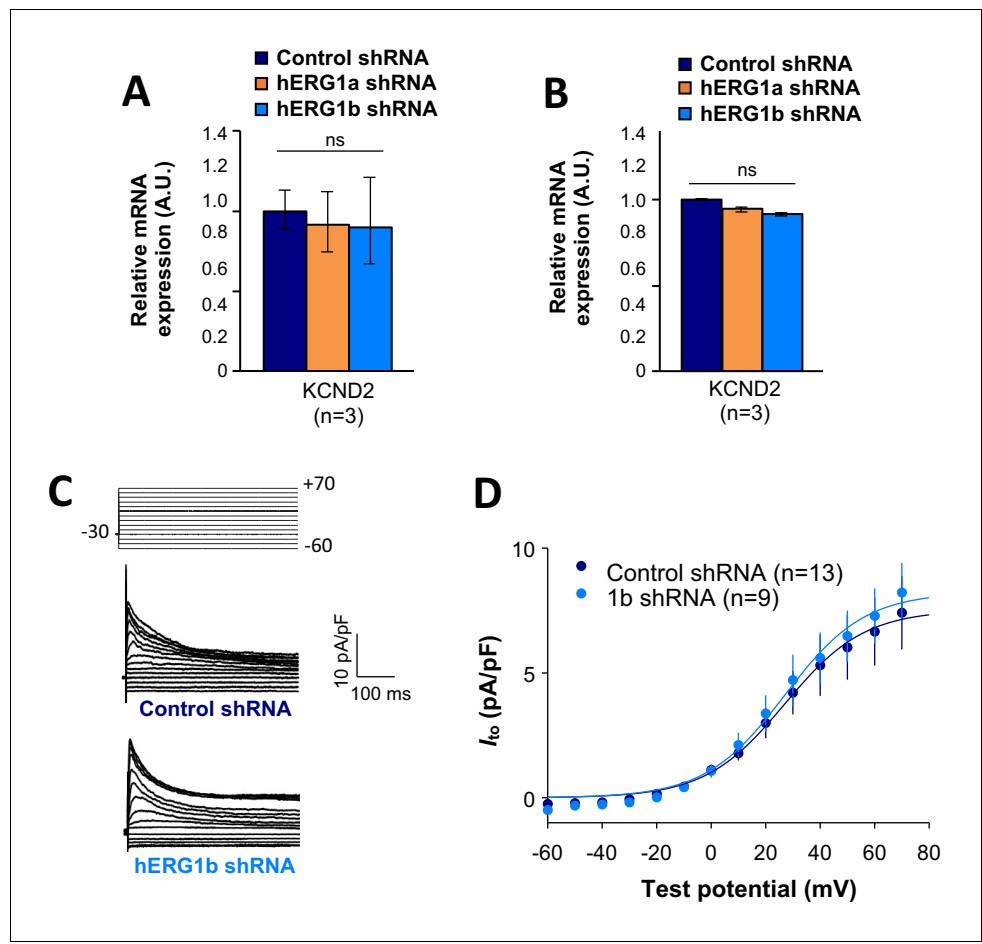

**Figure 8.** Effects of hERG1b silencing on $I_{to}$ and $K_V4.2$ channels in iPSC-CMs. (**A**) Effects of *hERG1a* or *hERG1b* silencing on $K_V4.2$ channel mRNA expression levels detected by RT-qPCR (mean ±95% CI) in IPSC-CMs. A non-targeting shRNA (scrambled shRNA) is used as a control. (**B**) Effects of specific *hERG1a* or *hERG1b* silencing on $K_V4.2$ channel mRNAs expressed alone in HEK293 cells. (**C**) Representative family of traces show $I_{to}$ in presence of control (upper) or hERG1b shRNA (lower). (**D**) Summary of steady-state current density vs. test potential shows effect of hERG1b shRNA (mean ±SE).

DOI: https://doi.org/10.7554/eLife.52654.041

The following source data is available for figure 8:

**Source data 1.** KCND2 is not affected by the co-knockdown effect raw data for *Figure 8A–B*.
DOI: https://doi.org/10.7554/eLife.52654.042

**Source data 2.** *Ito* current is not affected by the silencing of hERG raw data for *Figure 8D*.
DOI: https://doi.org/10.7554/eLife.52654.043

scrambled shRNA as a control. Cells were then collected for experiments 48 hr after last transfection.

## Antibodies

Rabbit anti-hERG1a (#12889 from Cell Signaling, 1:100), rabbit anti-hERG1b (#ALX-215–051 from Enzo, 1:100), rabbit anti-pan hERG (#ALX-215–049 from Enzo, 1:3000), rabbit anti $Na_V1.5$ (#ASC-005 from Alomone or #D9J7S from Cell signaling, 1:500), were used for immunofluorescence, western blot or RNA-IP experiments. Alexa 647 goat anti-rabbit, Alexa 488 goat anti-rabbit or Alexa 488 donkey anti-mouse were employed for indirect immunofluorescence or immunoblotting experiments (Thermofisher; 1:1000).

## RNA isolation and semi-quantitative real-time PCR

RNA isolation and purification were achieved using TriZol reagent (Life Technologies) and RNeasy Mini Kit (Qiagen). RT-qPCR experiments were performed using a TaqMan Gene Expression Assay (Life Technologies) and mRNA expression levels were calculated using the $2^{-\Delta\Delta Ct}$ cycle threshold method. All data were normalized to mRNA level of $\beta$-actin housekeeping genes. Because iPSC-CMs are subject to inherent biological variability, we used a standardization procedure to normalize the independent biological replicates as previously described (*Willems et al., 2008*). Briefly, a log transformation of the normalized relative expression gene level was performed, followed by mean centering and autoscaling of the data set. Results are expressed as average and 95% confidence intervals. Primers were purchased from Invitrogen (*hERG1a*: Hs00165120_m1; *hERG1b*: Hs04234675_m1; *SCN5A*: Hs00165693_m1; *RYR2*: Hs00892883_m1; and *β-actin*: Hs01060665_g1).

## Immunofluorescence

For immunofluorescence studies, iPSC-CMs were grown on gelatin-coated coverslips, rinsed in PBS three times and fixed in 4% paraformaldehyde for 10 min at room temperature. Following fixation, cells were incubated 1 hr at room temperature with a solution containing 0.5% triton X-100 for permeabilization and 1% bovine serum albumin along with 10% serum (secondary antibodies species) diluted in PBS to saturate samples and limit nonspecific binding. Cells were then processed for indirect immunofluorescence using a combination of primary and secondary antibodies (see antibodies section above). Cells were washed three times with PBS, incubated with DAPI to counterstain nuclei and mounted with Vectafield mounting medium.

## Single-molecule fluorescence in situ hybridization (smFISH)

FISH was performed using Stellaris probe sets, which comprised up to 48 oligonucleotides designed to selectively bind in series the targeted transcripts. Probes were designed using the StellarisTM Probe Designer by LGC Biosearch Technologies with the following parameters: masking level: 5, oligo length: 20 nucleotides, and minimum spacing length: two nucleotides. Oligonucleotides were labeled with TAMRA or Quasar 670 dyes for detection of *SCN5A* and *hERG* respectively. 48 oligonucleotides were designed for *SCN5A*, *RyR2* and *GAPDH* and 35 for the specific N-terminal sequence of *hERG1a*. Sequences for all probes are provided in Supplementary Table 1. FISH was performed on iPSC-CMs according the manufacturer's protocol. Briefly, fixation was performed by adding paraformaldehyde to a final concentration of 4% (32% solution, EM grade; Electron Microscopy Science) followed by a hybridization step for at least 4 hr at 37°C in a buffer containing a final concentration of 125 nM probes and 10% formamide (Stellaris hybridization buffer). Cells were washed for 30 min (Stellaris washing buffer A) before incubation for 30 min at 37°C with DAPI to counterstain the nuclei. A final washing step was performed (Stellaris washing buffer B) and coverglasses were mounted onto the slide with Vectashield mounting medium.

Digital images were acquired using a 63X objective on a Leica DMi8 AFC Inverted wide-field fluorescence microscope. Z-sections were acquired at 200 nm intervals. Image pixel size: XY, 106.3 nm. Image post-treatments were performed using ImageJ software (NIH). Briefly, a maximum projection was performed before background subtraction and images were filtered using a Gaussian blur filter to improve the signal/noise ratio and facilitate spot detection. Spot detection and colocalization was performed using the plugin ComDet on ImageJ (*Chang et al., 2006*; *Hoffman et al., 2001*).

FISHQUANT was used as a second method for spot detection and gave similar values. Briefly, background was substracted using a Laplacian of Gaussian (LoG) and spots were fit to a three-dimensional (3D) Gaussian to determine the coordinates of the mRNA molecules. Intensity and width of the 3D Gaussian were thresholded to exclude non-specific signal (*Raj et al., 2008*; *Femino et al., 1998*).

To evaluate the number of mRNA molecules, the total fluorescence intensity of smFISH signals was fitted with the sum of Gaussian functions (see equation below) to determine the mean intensity of a single mRNA.

$$y = y_0 + \frac{A}{w\sqrt{\frac{\pi}{2}}}\, e^{-2\left(\frac{x-xc}{w}\right)^2}$$

## Statistical analysis of smFISH and IF

For the purpose of our statistical calculations, we assumed that the protein and mRNA signals were circular. The following formulas were used to calculate the expected number of mRNAs ($E_m$) that would interact based on chance alone for each association criteria:

$$E_m = \frac{N_{m1}N_{m2}(2\pi r^2 - I)}{A}$$

where $N_{m1}$ is the total number of mRNA in one channel, $N_{m2}$ is the total number of mRNA in the second channel, r is the average radius of mRNA spots (in nm), I is the intersection between particles (nm$^2$, and A is the total area of the region analyzed (in nm$^2$. As the distance between particles is increased, the number of expected associated mRNAs will increase since more mRNAs will be considered associated. We used criteria with different stringency in the first set of experiments (from 1 pixel to four pixels distance between spots) and considered the two pixels distance between spots physiologically relevant for triple association analysis and co-knockdown experiments.

To test the significance of triple associations between hERG1a mRNA, SCN5A mRNA and hERG1a protein, the following formula was used:

$$E_p = \frac{N_p E_m(\pi r^2 - I)}{A}$$

where $N_p$ is the total number of proteins, $E_m$ is the expected number of mRNA that would interact based on chance alone as calculated above. For each association criteria, the intersection between particles was calculated using the following equation:

$$I = 2r^2 \cos^{-1}\left(\frac{d}{2r}\right) - \frac{1}{2}d\left(\sqrt{4r^2 - d^2}\right)$$

## Correlation analysis

mRNA numbers were plotted against each other from different combinations of smFISH signals as scatter plots. Then Pearson's and Spearman's correlation coefficients were evaluated to assess correlation between considered mRNA species.

The following equation was used to calculate Pearson's coefficient R and determine the coefficient of determination R$^2$ from the mRNA pairs $x_i$, $y_i$:

$$R = \frac{Cov(x_i, y_i)}{\sigma_{x_i} - \sigma_{y_i}}$$

where $Cov(X_i, Y_i)$ is the covariance of the values and $\sigma_{x_i} - \sigma_{y_i}$ is the difference between the standard deviation of the values. Significance was determine using a F test.

The Spearman's coefficient ρ was determined on ranked values X$_i$ and Y$_i$ using the following equation:

$$\rho = \frac{Cov(X_i, Y_i)}{\sigma_{X_i} - \sigma_{Y_i}}$$

where $Cov(X_i, Y_i)$ is the covariance of the rank values and $\sigma_{X_i} - \sigma_{Y_i}$ is the difference between the standard deviation of the ranked values. Significance was determine using two-tailed probability test.

## RNA-IP (RNA-immunoprecipitation)

Ribonucleoprotein (RNP) complexes were isolated with a RiboCluster Profiler TM RIP-Assay Kit (Medical and Biological Sciences) using protein-specific antibodies and Ab-immobilized A/G agarose beads. After formation of the RNP/beads complex, we used guanidine hydrochloride solution to dissociate beads from RNP complexes. Finally, target RNAs were analyzed using RT-PCR.

## Electrophysiological measurements

Patch clamp under whole-cell configuration was used to record all ionic currents. $I_{Kr}$ and $I_{Na,late}$ were recorded at physiological temperatures (37°C), while $I_{Na}$ was recorded at room temperature (22°C)

using an Axon 200B amplifier and Clampex Software (Molecular Devices). Glass pipettes with a resistance of 2.5–5 MΩ measured with physiological solutions (below) were pulled using an automatic P-97 Micropipette Puller system (Sutter Instruments).

To record steady state and tail $I_{Kr}$, cells were continuously perfused with an external solution containing (in mM): NaCl 150, KCl 5.4, CaCl$_2$ 1.8, MgCl$_2$ 1, Glucose 15, HEPES 15, Na-pyruvate 1, and the pH was adjusted to 7.4 with NaOH. Pipettes were filled with an internal solution containing (in mM): NaCl 5, KCl 150, CaCl$_2$ 2, EGTA 5, HEPES 10, Mg-ATP 5, and the pH was adjusted to 7.3 with NaOH. The voltage protocol for $I_{Kr}$ was completed at physiological temperature (37°C) and determined as an E-4031 (2 µM) sensitive current. Cells were recorded using a holding potential of −50 mV, followed by a pulse at −40 mV to inactivate sodium channels, then 3 s depolarizing steps (from −50 to +30 mV in 10 mV increments) to activate hERG channels and finally to −40 mV for 6 s. Steady-state $I_{Kr}$ was measured as the 5 ms average current at the end of the depolarizing steps. Tail currents were measured following the return to −40 mV.

To record $I_{Na}$, cells were perfused with an external solution containing (in mM): NaCl 50, Tetraethylammonium (TEA) methanesulfonate 90, CaCl$_2$ 2, MgCl$_2$ 1, Glucose 10, HEPES 10, Na-pyruvate 1, Nifedipine 10 µM, and pH adjusted to 7.4 with TEA-OH. Micropipettes were filled with an internal solution containing (in mM): NaCl 10, CaCl$_2$ 2, CsCl 135, EGTA 5, HEPES 10, Mg-ATP 5, and pH was adjusted to 7.3 with CsOH.

$I_{Na}$ activation was investigated by applying pulses between −140 and +20 mV in 10 mV increments from a holding potential of −120 mV. To measure inactivation of sodium channels, conditioning pulses from −140 to +20 mV in 10 mV increments were applied from a holding potential of −120 mV following by a test pulse to −20 mV.

To record $I_{Na,late}$, cells were perfused with an external solution containing (in mM): NaCl 140, CsCl 5.4, CaCl$_2$ 1.8, MgCl$_2$ 2, HEPES 5, Nifedipine 10 µM, and pH was adjusted to 7.3 with NaOH. Pipette were filled with an internal solution containing (in mM): NaCl 5, CsCl 133, Mg-ATP 2, TEA 20, EGTA 10, HEPES 5, and pH was adjusted to 7.33 with CsOH. $I_{Na,late}$ was measured by applying an 800 ms single pulse to −30 mV from a holding potential of −120 mV. Late $I_{Na}$ was measured as the current integral from 50 to 800 ms from the beginning of the pulse.

To record $I_{to}$, cells were continuously perfused with an external solution containing (in mM): NaCl 150, KCl 5.4, CaCl$_2$ 1.8, MgCl$_2$ 1, Glucose 15, HEPES 15, Na-pyruvate 1, E4031 2, CdCl$_2$ 0.5 and the pH was adjusted to 7.4 with NaOH. Pipettes were filled with an internal solution containing (in mM): NaCl 5, KCl 150, CaCl$_2$ 2, EGTA 5, HEPES 10, Mg-ATP 5, and the pH was adjusted to 7.3 with NaOH.

Both activation (for $I_{Kr}$, Ito and $I_{Na}$) and inactivation (for $I_{Na}$) were fitted to Boltzmann equations (*Equations (1) and (2)*, respectively) and voltage dependence parameters were obtained.

$$I(V) = \frac{(V - V_{rev})G_{max}}{1 + e^{\frac{(V - V_{1/2})}{k}}} \tag{1}$$

$$I(V) = \frac{(I_{min} - I_{max}) + I_{max}}{1 + e^{\frac{(V - V_{max})}{k}}} \tag{2}$$

## Data availability

The source data corresponding to *Figures 1B*, *2B, C, D, E*, *3B, D*, *4B, C, E*, *5N, C, D*, *6N, C, D*, *7A, N, D, E, G, H, J, D*, *8A, B and D* are provided.

## Acknowledgements

Research reported in this publication was supported by the National Heart, Lung and Blood Institute of the National Institutes of Health R01HL131403. The authors thank Dr. Peter Mohler of the Dorothy Davis Heart and Lung Institute for heart samples and Drs. Barry Ganetzky of the University of Wisconsin-Madison, Andrew Harris of the Rutgers New Jersey Medical School and Drs. Cynthia Czajkowski and Baron Chanda of the University of Wisconsin School of Medicine and Public Health for comments on an earlier version of the manuscript.

## Additional information

### Funding

| Funder | Grant reference number | Author |
| --- | --- | --- |
| National Heart, Lung, and Blood Institute | 1R01HL131403-01A1 | Gail A Robertson |
| National Heart, Lung, and Blood Institute | 5T32HL007936-01A1 | Erick B Rios-Pérez Jennifer J Knickelbine |

The funders had no role in study design, data collection and interpretation, or the decision to submit the work for publication.

### Author contributions

Catherine A Eichel, Conceptualization, Data curation, Formal analysis, Investigation, Methodology, Writing—original draft, Writing—review and editing; Erick B Ríos-Pérez, Fang Liu, Conceptualization, Data curation, Formal analysis, Methodology; Margaret B Jameson, David K Jones, Jennifer J Knickelbine, Data curation, Formal analysis; Gail A Robertson, Conceptualization, Supervision, Funding acquisition, Writing—original draft, Project administration, Writing—review and editing

### Author ORCIDs

Erick B Ríos-Pérez https://orcid.org/0000-0002-2246-9112
Margaret B Jameson https://orcid.org/0000-0003-1225-9194
Jennifer J Knickelbine https://orcid.org/0000-0003-3070-9128
Gail A Robertson https://orcid.org/0000-0003-4694-5790

### Decision letter and Author response

Decision letter https://doi.org/10.7554/eLife.52654.051
Author response https://doi.org/10.7554/eLife.52654.052

## Additional files

### Supplementary files

• Supplementary file 1. List of probes used in smFISH experiments. The probes were designed using Stellaris probe Designer software with the following parameters: 18 to 20 nucleotides oligo length, a masking level of 5, a minimum spacing length of 2 nucleotides and a maximum number of probes of 48. Due to the length of the N-terminal specific sequence for *hERG1a* mRNA, the number of probes used to detect *hERG1a* is limited to 35.
DOI: https://doi.org/10.7554/eLife.52654.044

• Supplementary file 2. Summary of correlation analysis perfomed in iPSC-CMs. The linear correlation between the different combination of mRNAs was evaluated using the Pearson correlation coefficient. Because the Pearson coefficient is highly sensitive to outliers and only assess linear correlation, the Spearman's correlation coefficient was also calculated. Both tests revealed a significant correlation between hERG1a and SCN5A mRNAs and no significant correlation for *hERG1a/RyR2*, *hERG1a/ GAPDH* and *SCN5A/GAPDH* pairs. Levels of significance were adjust with a Bonferroni correction taking into account correlation coefficients and either linear correlation or non-linear correlation for Pearson's and Spearman's test respectively.
DOI: https://doi.org/10.7554/eLife.52654.045

• Supplementary file 3. Summary of colocalization analysis perfomed in iPSC-CMs for different association criteria. Comparison of the average number of mRNAs particles observed to be associated and the expected number based on chance alone using centroid positions and different association criteria (from touching to 67% overlap). The significance is tested with a paired t-test Bonferroni's correction. The number of *hERG1a* and *SCN5A* mRNAs observed to be associated is significantly

above that expected by chance alone for all association criteria tested while no significant differences are observed for *hERG1a/RyR2*, *hERG1a/GAPDH* and *SCN5A/GAPDH* associations.
DOI: https://doi.org/10.7554/eLife.52654.046

• Supplementary file 4. Voltage dependence of activation and inactivation parameters for the sodium channels in cells transfected with a control shRNA or a hERG1b specific shRNA. Parameters were obtained after fitting to a Boltzmann equation activation and inactivation data.
DOI: https://doi.org/10.7554/eLife.52654.047

• Supplementary file 5. Voltage dependence of activation of hERG channels in cells transfected with a control shRNA or a hERG1b specific shRNA. Parameters were obtained by fitting the experimental data of the I-V curve of the peak tail $I_{Kr}$ to a Boltzmann equation.
DOI: https://doi.org/10.7554/eLife.52654.048

• Transparent reporting form DOI: https://doi.org/10.7554/eLife.52654.049

### Data availability

All data generated or analysed during this study are included in the manuscript and supporting files. Source data files have been provided for all figures.

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
