## [Decision Letter]

**Acceptance summary:**

The paper demonstrates a regulatory mechanism for maintaining the homeostatic "balance" between sodium and potassium channels in heart tissue, where imbalance can lead to dangerous arrhythmias. Transcripts of ERG potassium and Na_v_1.5 sodium channels are colocalized and cotranslated, which helps maintain a safe ratio of the depolarizing sodium and the hyperpolarizing potassium channels.

**Decision letter after peer review:**

The previous reviews from another journal have been considered and evaluated along with the manuscript by an expert Reviewing Editor, who has the following comments:

"On one hand, I think the reviewers did an excellent job and identified shortcomings preventing the ability to come to clear and definite conclusions, but on the other hand, the work, as one of the reviewers noted, is thought provoking and as such does it really need to dot every “i” and cross every “t”? I think if the authors would point out in the Discussion that for technical reasons, they were unable to do the RNAseq and similarly for the other really technically difficult points, it would be okay. They are being very honest in what they have confidence in and what not, more than I can say for many other papers. So, I would accept it with minor revisions, especially in discussion to qualify their conclusions, and let people make up their own minds. It is fun to read something a little bit different."

---

## [Author Response]

We have submitted our revised manuscript entitled “A microtranslatome coordinately regulates sodium and potassium currents in the heart” for review at *eLife*. We have provided reviews and responses from another journal to assist in *eLife’s* consideration of our manuscript for publication.

This manuscript describes for the first time the cotranslational association and coregulation of mRNA transcripts encoding different ion channels species. The balance of ion currents that generate electrical impulse is essential for proper function in excitable cells. The observation that transcripts associate and are coregulated via a “microtranslatome” may represent a general mechanism by which cells coordinate expression and thus activity of functionally related proteins. Thus the primary audience will be those interested in a new mechanism by which multiple conductances are coordinately regulated at the level of protein translation. We received the editorial suggestion from Dr Aldrich to acknowledge the difficulty of conducting RNA-seq analysis as part of this study and have now revised the manuscript to include the following text in the Discussion: “A comprehensive analysis of the microtranslatome’s components will require RNA-seq at a level of multiplexing that ensures sufficient statistical power in the face of potentially reduced complexity of the RNA-IP samples. These efforts will necessarily be followed by validation through complementary approaches such as RNAi and smFISH to confirm their identity within the microtranslatome.”

[Editors' note: we include below the reviews that the authors received from another journal, along with the authors’ responses.]

We appreciate the opportunity to improve our manuscript per the critiques of the reviewers and editor. A major concern by the reviewers and reiterated by the editor was whether the association of transcripts was cotranslational. As described below we have carried out experiments using puromycin showing dissociation of the complexes and providing a third line of support for the cotranslational basis of *hERG1a* and *SCN5A* transcript association. Despite our best attempts not all the suggestions were fulfilled, as detailed below, but we hope you agree that the manuscript has been significantly improved as a result of the key experiments completed.

Editorial comments:1) Please note that we do not think that identifying proteins within the 'translatome' through mass spectrometry as suggested by reviewer 1 (point 2) is necessary. However we do feel that attempting to identify the RNA content through non-biased sequencing (RIP-Seq or similar approach rather than just targeted PCR) would significantly increase confidence in the original result. Given that the conditions for immunoprecipitation have already been worked out, this should be quite feasible within the timeframe of a regular revision, and we therefore consider this demand within the scope of the current manuscript.

RNA-seq analysis has been a vexing aspect of this project. We carried out a detailed study that produced confusing (and expensive) results. Upon attempting to repeat the study, a new campus core with greater bioinformatics and study design expertise than the company we originally used suggested that our results were likely hampered by random variation that arises with libraries of reduced complexity, such as those expected of our microtranslatome complex. To carry out the experiment again will require many more replicates to mitigate this problem and will be extremely expensive. We believe that with the addition of the puromycin perturbation our results demonstrating the presence of the hERG mRNA in the complex via RT-PCR, smFISH, and co-knockdown are now quite compelling.

2) Addressing reviewer 1's point 4 experimentally will also be necessary.

We agree and please see our response under reviewer 1, point 4.

3) In regard to reviewer 2's comments, we agree that several of the experiments lack adequate controls, which should be remediated. We also agree with this reviewer that evidence the association occurs co-translationally is currently lacking and must be provided.

As described below, we have carried out experiments with puromycin to disrupt the interaction between the ribosomes and mRNA, and report in Figure 6 the results that the antibody and smFISH signals are dissociated as would be expected from a cotranslational complex.

Reviewer #1:

This is a well-written, thought-provoking and novel study with potential for significant impact to the field. I only have a few comments.1) It seems to me that with not much additional effort the investigators can describe more than two transcripts in these complexes. Could it be possible to run an RNA-seq on the precipitates to know what else, in addition to the two transcripts described, is present?

We have identified other elements in the precipitate using a candidate approach and are sorting out whether they are components of the same or different microtranslatomes. This is a challenging task that requires multiplexed co-knockdowns and smFISH experiments with each component. We are currently approaching this challenge as a different project that is the thesis project of one of my students, who has just begun these experiments. Also, as mentioned above, RNA-seq of the microtranslatome, which we attempted, produced confusing results and will require a more sophisticated and expensive approach than is required with, e.g., a whole-cell or tissue sample. Even if we did obtain a list of other interacting mRNAs we hope the reviewer will agree that the level of work required to prioritize the candidates and validate their association with the complex will require much work beyond the scope of the current project.

2) Similar to the comment above: A proteomic analysis of these precipitates would identify other proteins whose transcripts may also be present (as per the RNA-seq experiment suggested above) as well as (and perhaps more importantly) which known RNA-binding proteins are present in the complex that may be acting to form the complex.

We agree with the reviewer that the results from a proteomic analysis will be extremely important, but as the editor suggested, this experiment is beyond the scope of the current study. Indeed, this is currently the sole project of a postdoctoral fellow in the lab. Again, validation of each protein component, such as RNA binding proteins, will require a tremendous effort to demonstrate their functional roles.

3) And from both points above: I am not too keen on the term “excitotranslatome.” It unnecessarily limits the relevance of the findings to the context of electrical excitability. These complexes may cover more than just one function. Yet again, this is only an opinion (the authors discovered them so of course, they should be free to name them…)

We agree with this viewpoint and now term the complex a “microtranslatome.” We have changed the title accordingly to: “A microtranslatome coordinately regulates sodium and potassium currents in the heart.”

4) Some of the speculation in the Discussion section would presume that the co-translation also leads to equivalent amounts of functional channels. It would be interesting to prove the latter. Do current amplitudes correlate on a cell-by-cell bases? Are these proteins (HERG and Na_v_1.5) co-localizing at the cell membrane? If you were to do a macropatch of the cell surface, would you be able to record both channels under the same small patch?

Whether the channel expression is quantitatively correlated is an important test of our central hypothesis and we thank the reviewer for this suggestion. We recorded *I*_Kr_ and *I*_Na_ current amplitudes simultaneously in the same cell as shown in Author response image 1. We did not uncover a strong correlation between *I*_Kr_ and peak *I*_Na_ current densities on a cell-by-cell basis (R^2^=0.28), but we have little confidence in the outcome of this experiment because I_Kr_ is extremely small relative to the noise and thus difficult to reliably quantify on a cell-by-cell basis. In the future we hope to quantify these currents in native, mature cardiomyocytes in which the currents are larger and correlations between *I*_Kr_ and late *I*_Na_ can be made.

**Author response image 1. respfig1:** Correlation analysis of hERG and sodium currents. (**A**) Voltage step protocol used to measure both sodium (*I*_Na_) and hERG (*I*_Kr_) currents in the same cell, using cesium as the charge carrier for hERG channels. The region from where *I*_Kr_ was measured is highlighted in the protocol (yellow square). Representative traces before (Control) and after perfusing E-4031 (10 µM) to block hERG channels, validating that the recorded current is *I*_Kr_. (**B**) Current density of hERG channels were plotted in function of Na_v_1.5 current density and correlation coefficients are shown on top of the graph. One cell (red dot) was excluded from the correlation analysis as it meets the outlier criteria. n = 20 cells of 2 independent experiments (N = 2).

The assessment of protein colocalization at the cell membrane would have been an interesting complement to confirm this correlation. Unfortunately, despite much effort put towards identifying antibodies produced in different species required for colocalization in immunofluorescence experiments, none gave sufficiently specific signals for hERG or Na_V_1.5 proteins. Work along this line continues in the lab.

Macropatch would be an elegant approach to identify a functional proximity of I_Kr_ and I_Na_ channels. Unfortunately, while sodium current is big enough to be recorded in macropatch configuration in iPSC-CMs, the same is not true for *I*_Kr_, which is only 4-5 pA/pF in whole-cell configuration.

Other minor comments:Figure 1: Isolated examples are presented. A measure of reproducibility should be provided. This applies particularly to the human LV and the iPSC-CM samples: How many different hearts investigated? How many clones of hIPSC-CMs? How many repeats?

Thank you for noticing this oversight. We purchased iPSC-CMs from Cellular Dynamics International (CDI) and those cells are derived from a single clone. They are reliable and consistent between batches, but multiple clones are unavailable. We now state the N for independent experiments for iPSC-CMs and human LV in the Figure 1 legend.

“In a series of 35 or 48…” What does the 35 or 48 refer to?

These values refer to the number of probes in sequence used for the smFISH experiments. These values have been removed from the text and clarified in the Materials and methods to avoid confusion.

Figures 2-3: Same as Figure 1: number of repeats? N values?

The n values have been added for each experiment directly on the graph or in the legend for each figure.

Figure 3: It would be interesting to see the complementary plot: SCN5A versus RyR2.

These probes were ordered with the same fluorophore in the original study and were therefore not distinguishable under the microscope. Unfortunately, we overlooked this suggestion until very recently and to avoid further delay we hope that the reviewer will agree that although this is interesting, it is not absolutely necessary to complete the study.

Figures 4B and 4E: Are the units in the X axis correct (micrometers)?

Thank you for this careful observation. This error has been corrected in the figure.

Reviewer #2:

In this manuscript, the authors study the co-translational assembly of mRNAs that encode two different ion channels (I_Na_ and I_Kr_) that are encoded by SCN5A and hERG in iPSC cardio myocytes. The association of the two mRNAs is first shown by RIP experiments and then further evaluated by smFISH and smFISH^+^IF for the hERG1a protein. Interestingly, RNAi knockdown of hERG1 mRNA also results in a depletion of SCN51 mRNA suggesting a connection between the stabilities of their transcripts by their association. The authors also show test the physiological consequence of this co-depletion by electrophysiology measurements.In general, the work is interesting and addresses an important question in the field. The work would benefit from several controls in order to strengthen their assertion that these affects are occurring via co-translation.

The authors thank the reviewer for the expression of enthusiasm regarding the work’s significance. We recognize the primary concern is whether the phenomena and effects observed are cotranslational and will address this concern among the other comments as well.

1) Figure 1B is very dense and difficult to interpret. It would be better to separate out results from iPSC-CMs and HEK ectopic-expression. It is also not clear why RyR2 was also not overexpressed in HEK cells. The negative data for KCNJ2 can probably be moved to supplemental as well as IgG control data. Are these interactions puromycin-sensitive? I think this is a key control that is missing throughout the manuscript and should be performed for all of the smFISH studies as well.

We have revised the gel per the reviewer’s suggestion and included the previous, more complete gel in the supplement for those wanting to see all the controls within a single experiment. The RYR2 was not expressed heterologously because we were unable to express the available construct and therefore relied on natively expressed RYR2 in heart and iPSC-CMs, but we did not feel this affected the conclusion that the RYR2 is not a component of the microtranslatome. Having a negative control (KCNJ2) throughout the experiment is important to show that the IP is specific, so we kept this in Figure 1.

We have carried out experiments using puromycin and now show in Figure 6 that the association of *hERG1a* and *SCN5A* mRNAs with hERG protein is robustly reduced by the treatment, providing another line of support for the cotranslational association of the transcripts.

2) It is not clear what is the importance of these clusters as they are not mentioned again? Are the clustered puro-sensitive? Do they require active translation to form or is that just clustering on the ER somehow? Have the authors considered mild-digitonin extraction to remove cytosolic mRNAs and to leave behind ER- localized ones as described here (PMID: 23271194)?

Please see response to point 4 below.

3) GAPDH is not a good control for their experiments since it encodes a cytosolic protein that does not translate to high levels on the ER. The RyR2 transcript is much better since it encodes also encodes a membrane protein. I would either remove all GAPDH data or move to supplemental since its comparison is almost meaningless for what they want to show.

We value the information provided by the GAPDH signal precisely because it represents a transcript with a completely different type of distribution; not only is it distributed differently from the channel transcripts, but it is also abundant, in contrast to the rarer channel transcripts. In addition, the GAPDH probes are validated by Stellaris, adding another measure of certainty regarding signal identification.

4) The authors show that about 25% of transcripts co-localize. Is this different for single or clustered transcripts?

We carried out an analysis of all transcripts expressed in cells and found no major change in distribution in clusters with puromycin treatment (Author response image 2). When considering colocalized transcripts specifically, we found that colocalized *hERG1a* and *SCN5A* mRNAs are preferentially found in pairs of single molecules (32% and 36% of *hERG1a* and *SCN5A* transcripts respectively) or organized in clusters of 6 or more molecules (27% and 23% for *hERG1a* and *SCN5A* mRNAs respectively; Author response image 2). Interestingly, puromycin treatment shifted these proportions towards association in clusters of 6 or more molecules suggesting that ribosomes are involved in the association of transcripts found as single molecule. We could speculate on the significance of this finding, but feel these observations raise many more questions than they answer and plan to use these findings as the basis for a new study. We now note in the manuscript that “Further work will be required to elucidate the significance or possible physiological role of differently sized mRNA clusters” so as to acknowledge the unanswered question.

**Author response image 2. respfig2:** Effect of puromycin on hERG and SCN5A cluster distribution. (**A**) The distribution of the number of mRNA molecules associated in clusters for each transcript evaluated by smFISH in presence or absence (control) of puromycin. (**B**) Histogram showing the distribution of colocalized particles in clusters as a fraction of total *hERG* colocalized particles with or without puromycin treatment. (**C**) Histogram showing the distribution of colocalized particles in clusters as a fraction of total *SCN5A* colocalized particles with or without puromycin treatment.

Given the similar transcripts numbers, I do not understand the different expected co-localization numbers in Supplementary file 2. I would assume that the other parameters are identical.

The reviewer is right about the numbers of transcripts being similar for *hERG* and *SCN5A* mRNAs, but the parameter being modified is how much overlap there is between colocalized particles. Thus, the 1 pixel distance between center of mass (67% of the two spots overlapping) will consider a smaller number of particles as colocalized than 4 pixels (when spots touch each other). We modified the Materials and methods in order to clarify this point.

5) As it is not clear if the protein detected is nascent on polysomes or mature, I think it is not very strong evidence that this occurring co-translational. Is this sensitive to puro?

Evidence that transcripts associate cotranslationally has been clarified in the manuscript and can be summarized as follows: 1) The co-purification of transcripts with protein in the RNA-IP experiments implying that they are undergoing translation; 2) the co-knockdown of transcripts undergoing translation (as shown by quantitative RT-PCR and patch clamp analysis of currents); and 3) new data provided in Figure 6 showing a reduction of association of transcripts with hERG1a protein in the presence of puromycin.

6) This analysis would be more interesting with smFISH. Are they degrading the mRNAs that co-localize or do not co-localize or both? This might help them distinguish between possible models for this co-depletion.

We carried out the co-knockdown experiment using smFISH as suggested by the reviewer and it is now presented in Figure 7—figure supplement 1 and described in the manuscript. We observed that numbers of *hERG1a* and *SCN5A* mRNAs are decreased upon hERG1b-specific silencing (Figure 7—figure supplement 1B) as is the number of colocalized particles (Figure 7—figure supplement 1C). This observation supports the existence of a mechanism that controls the relative quantity of those two transcripts, as stated in the manuscript.

Minor comments:1) Please define iPSC-CM the first time it is used.

This has been added to the manuscript.

2) Figure 4B and E – distance should be in nm.

This has been corrected.